# A Mathematical Radiobiological Model (MRM) to Predict Complex DNA Damage and Cell Survival for Ionizing Particle Radiations of Varying Quality

**DOI:** 10.3390/molecules26040840

**Published:** 2021-02-05

**Authors:** Spyridon A. Kalospyros, Zacharenia Nikitaki, Ioanna Kyriakou, Michael Kokkoris, Dimitris Emfietzoglou, Alexandros G. Georgakilas

**Affiliations:** 1Physics Department, School of Applied Mathematical and Physical Sciences, National Technical University of Athens (NTUA), 15780 Zografou, Greece; spkals@central.ntua.gr (S.A.K.); znikitaki@mail.ntua.gr (Z.N.); kokkoris@central.ntua.gr (M.K.); 2Medical Physics Lab, Department of Medicine, University of Ioannina, 45110 Ioannina, Greece; ikyriak@uoi.gr (I.K.); demfietz@uoi.gr (D.E.)

**Keywords:** mathematical radiobiological model (MRM), complex DNA damage, cell survival, relative biological effectiveness (RBE), high-LET

## Abstract

Predicting radiobiological effects is important in different areas of basic or clinical applications using ionizing radiation (IR); for example, towards optimizing radiation protection or radiation therapy protocols. In this case, we utilized as a basis the ‘MultiScale Approach (MSA)’ model and developed an integrated mathematical radiobiological model (MRM) with several modifications and improvements. Based on this new adaptation of the MSA model, we have predicted cell-specific levels of initial complex DNA damage and cell survival for irradiation with ^11^Β, ^12^C, ^14^Ν, ^16^Ο, ^20^Νe, ^40^Αr, ^28^Si and ^56^Fe ions by using only three input parameters (particle’s LET and two cell-specific parameters: the cross sectional area of each cell nucleus and its genome size). The model-predicted survival curves are in good agreement with the experimental ones. The particle Relative Biological Effectiveness (RBE) and Oxygen Enhancement Ratio (OER) are also calculated in a very satisfactory way. The proposed integrated MRM model (within current limitations) can be a useful tool for the assessment of radiation biological damage for ions used in hadron-beam radiation therapy or radiation protection purposes.

## 1. Introduction

Ionizing Radiation (IR) is considered to be a “double-edged sword” because it not only affects the basic biological properties of neoplastic cells but also restructures normal cells causing serious damage to them. Hadron radiation therapy has been extensively used during the last decades all around the world as a part of the treatment for oncological patients as a potential alternative to conventional radiation therapy [1]. In contrast to ionizing charged particles which have a finite range, ionizing photons (X- or γ-rays) are able to penetrate matter and deposit energy at all depths in a patient’s body, albeit with exponentially decreasing dose levels after the first few centimeters. The finite range of a charged particle is exploited in hadron therapy to reduce dose and therefore biological damage to healthy tissue in proximity to a tumor target. In particular, the rapid increase in the Linear Energy Transfer (LET) = *dE/dx* (*dE* represents the energy loss and *dx* represents the increment of path length) at the end of an ion’s path through a patient’s body gives rise to the so-called *Bragg peak*. Ion beams exhibit a strong increase in LET in the Bragg peak as compared with the entrance region and they biologically attain higher cell killing and enhanced mutagenesis and chromosomal instability [2]. In cancer radiotherapy, the physical and biological properties of ion beams are more favorable than those of the conventional photon beams. On the other hand, the relative biological effectiveness (RBE). values for different ions used in radiotherapies tend to increase with increasing LET up to a maximum value before decreasing with further increases in LET. The RBE of an IR is defined as the ratio of a dose of standard radiation (usually low-LET γ-rays) to the dose of test radiation to produce same biological effects. Because LET is proportional to the square of the particle’s charge, ions with atomic numbers Z greater than a proton (Z = 1) have the ability to deliver more dose at larger LET values than protons (or X-rays). This attribute of ions with Z > 1, along with the increase in LET in the Bragg peak region, creates opportunities to increase the RBE in a tumor target without a concurrent increase in the RBE of nearby healthy tissue, i.e., the therapeutic ratio tends to increase with increasing atomic number. 

The most critical target regarding biological effects of IR in cells is the DNA macromolecule. However, recent studies [3,4] have shown that extranuclear damage also contributes to many cellular endpoints, including reproductive cell death. IR produces a number of DNA lesions types, such as base damages or closely spaced (10–20 bp) clustered lesions formed through the direct deposition of energy in the DNA and by the indirect reaction with radicals and reactive oxygen species (ROS) that can diffuse a few nanometers away from the ion’s trajectory and damage the DNA [5]. For clinical applications of particle radiotherapy, it is important to develop predictive models in order to quantify the level of biological damage induced. In recent decades, such radiobiological models have been developed [6,7] to describe the response of cells to IR, which correlate irradiated cells’ reaction to the delivered dose and other parameters expressing cell sensitivity. These models take into account the secondary electrons produced by the interaction of ions with the biological matter, free radical diffusion away from the locus of their production, as well as the biological effects of the deposited dose in combination with the local number density of these secondary particles. The role of low energy electrons in causing biological damage has been previously emphasized [8,9].

At the level of individual cells, X-or γ-rays deposit their energy uniformly at all locations within a cell, whereas charged particles deliver highly localized energy deposits within a few nanometers of the particle trajectory. The concept of an “absorbed dose”, which is the expected value of the stochastic distribution of energy deposits formed in a target of interest divided by the mass of the target, does not fully account for biologically relevant differences in the small-scale spatial deposition of energy by low- and high-LET radiations. Despite the limitations of the “absorbed dose”, it remains the most widely used parameter for relating the deposition of energy in cells or tissue to specific biological endpoints, including reproductive cell survival.

The most common example of the aforementioned models is the linear-quadratic (LQ) survival model [10] which provides a simple relationship between cell survival and delivered dose. In this model, the fraction of surviving cells S is modeled using a linear-quadratic (LQ) function of radiation dose *d*:−ln S = α*d* + β*d*^2^(1)
where α, β are cell-specific coefficients that vary with the LET of an ion (and the effective LET associated with the secondary electrons produced by X-rays). According to the interpretation first introduced in 1972 by Kellerer and Rossi [10], it was supposed that a lethal event is caused by a single radiation track (the linear term α*d*) or by two independent radiation tracks (the quadratic term β*d*^2^). Chadwick and Leenhouts considered that the linear term expresses non-repairable double strand breaks (DSBs) and the quadratic term expresses a combination of two sublethal single strand breaks (SSBs) [11]. The LQ model is considered to be more valid for doses that are not much larger than the value of the α/β ratio for a cell type [12]. This model, despite being of an empirical nature, is considered to be the best-fitting one for the description of cell survival under irradiation and is widely used both experimentally and clinically. Other models such as the Padé linear quadratic (PLQ) model [13], the universal survival curve (USC) model [14] and a mechanistic formulation of a linear-quadratic-linear (LQL) model [15] contribute to its further clinical and experimental applications. The amorphous track structure model which was introduced by R. Katz in the mid-60 s for describing the response of biological systems to heavy ion irradiation [16,17,18] utilizes the concept of action cross section as the probability of targets in the radiation detector being activated to extract cell killing. At that time, the knowledge of electron tracks was used for defining the properties of the ion ones represented in two dimensions. The main goal of this model is to calculate the RBE. This model provides a well-defined method for directly predicting cellular survival in mixed ion and photon fields [19]. However, in this model the evaluated relation of the radial dose distribution with the survival probability of the cells was based on survival curves for X-rays. This model also explained the changing shape of survival curves and the variation of radiosensitivity with LET.

The Local Effect Model (LEM) [20,21] describes radiobiological effects based on amorphous track structure in link with dose response after X-ray irradiation. The chosen-for-irradiation cell is divided into a vast number of tiny voxels and a modified LQ model is applied for every voxel in order to assess the number of lesions produced in such a voxel. The total number of lesions is derived by summing up the local lesions while the final state of the cell is determined depending on the number of lesions. Especially, in the LEM IV version [22], the reproductive fate of a cell is attributed to the local density of initial DSBs and their mutual proximity within the nucleus. The DSBs are classified as either ‘isolated DSB’ or ‘clustered DSBs’, but there are no specific assumptions about biological interactions of DSBs to form lethal lesions. The parameters α and β used in the LEM are taken from X-ray irradiation data. This model takes into account the details of track structure in the nm-scale. It also makes use of Monte Carlo (MC) methods and numerical techniques to determine some quantities used by it [23]. On the other hand, the mechanisms of action operating on the LEM’s spatial scale are not explicitly modeled in some cases. Additionally, while the LEM IV tends to minimize the number of ad hoc adjustable parameters, it increases the computational complexity of the model [24].

The Microdosimetric-Κinetic Model (MKM) [25] uses estimations of stochastic energy deposition into volumes of the µm-scale. This model is formulated in terms of potentially lethal lesions (PLL). It discerns two types of lesions, the type I which is not repairable and thus lethal to the cell, and the type II which may undergo one of four transformations. In the MKM, the cell nucleus is divided into domains so that PLL are limited to the domain in which they are created. The MKM uses six input parameters (three of them are cell-specific) and it assumes that it is the frequency of intra-track and inter-track pairwise PLL interactions which determines the formation of chromosome aberrations and cell death as a function of the LET of each particle. This model relies on the LQ model for the representation of low- and high-LET cell survival response [26].

The Repair-Misrepair-Fixation (RMF) model [24,27] links the induction of initial DSBs to the formation of lethal point mutations and chromosome aberrations. A coupled system of non-linear ordinary differential equations is used to model the time-dependent kinetics of DSB induction, rejoining and interaction to form lethal or non-lethal chromosome aberrations. It is assumed that it is the formation of lethal point mutations and chromosome damage rather than the initial DSBs or DNA lesion complexity which is the leading mechanism underlying the effects of particle LET on cell survival. The RMF model treats initial DSB formation as a compound Poisson process. This model makes use of the fast Monte Carlo Damage Simulation (MCDS) code [28,29] to determine the RBE for DSB induction. 

The concept of the lethal and potentially lethal lesions used by the latter models is promptly derived from the interpretation attempts of the LQ model. On the other hand, the real nature of these sublesions remains undefined until nowadays. These models consider the misrepaired lesions as lethal ones; on the contrary, according to new advances in radiobiology, it is proved that misrepaired lesions are involved in genomic instability and cell transformation rather than radiosensitivity and cell death or apoptosis [30]. These models are based on the notion of DNA repair, and the repair rate per unit time used by them is supposed to be constant; this is not always compatible with the repair kinetics which show the existence of a continuous DSB spectrum of repair efficiency probabilities [31]. Additional radiobiological questions to be solved include: (i) the proved hypersensitivity to low doses [32] and (ii) the fact that there are radiosensitive syndromes which are not caused by a decrease in DSB repair kinetics (as in the Huntington’s disease [33] in neurofibromatosis and Usher’s syndrome [34]).

The general mechanistic model proposed by McMahon et al. [35] predicts the radiosensitivity of cells irradiated by X-rays, proton and carbon ions with extension to other charged particle exposures, and calculates the RBE of carbon ions. It also obtains survival curves and corresponding LQ parameters for different cell lines. This model compares its predictions of radiosensitivity with 800 survival curves from the literature. It is based on the mechanistic model by the same authors published in 2016 [36] which begins from spatial distributions of DSBs, incorporates the kinetics of different DNA repair processes and the probability and severity of misrepair. It provides predictions of a range of endpoints including DNA repair, genetic aberration and cell survival based on a set of 11 fitting parameters describing different processes common across a range of cell types. Then, cell-specific predictions are made from these parameters based on specific phenotypic characteristics. The latest model proposed by Mc Mahon et al. extends the first one and generates predictions of X-ray sensitivity based on parameters established in the mechanistic fit, and then is combined with a Monte Carlo simulation of energy distribution around particle tracks, involving the fitting of an additional parameter to link energy distribution to DSBs. Τhe mechanistic model BIANCA (BIophysical ANalysis of Cell death and chromosome Aberrations) in addition to cell survival curves predicts chromosome aberrations, in the form of chromosome aberration dose-response curves which are considered as an important indicator of tissue damage [37,38]. It assumes that (1) IR produces ‘cluster lesions’, where each of them produces two independent chromosome fragments, (2) fragment mis-rejoining or un-rejoining generates chromosome aberrations, and (3) some of these aberrations may cause cell death. This model uses only two parameters, but several mechanisms are not described explicitly to avoid the introduction of further parameters. It also makes some approximations (e.g., two chromosome free-ends with initial distance smaller than a cut-off value will undergo end-joining with 100% probability, whereas two fragments with initial distance larger than that value will never undergo end-joining) which deflect from our knowledge of chromosome exchanges. It also accepts that cell death can result only from chromosome aberrations, bypassing other pathways such as apoptosis and necrosis. The model’s results of cell survival show a tendency to underestimate the surviving fraction at the higher doses. 

The mechanistic model of cellular survival proposed by Wang et al. [39] offers predictions between the DSBs induced because of the irradiation of a cell and the probability of its survival, as well as calculation of the RBE of charged particles. It uses two input parameters, of which one is the average number of DSBs yielded by each particle, and six fitting parameters. This model is based on the assumption that DSBs are the initial DNA lesions after cell irradiation and that primary particles which caused no DSBs do not contribute to cell death. It also considers that only the non-homologous end-joining (NHEJ) is the dominating pathway of DSB repair, and thus this contributes to cell death.

The RITCARD (Radiation Induced Tracks, Chromosome Aberrations, Repair, and Damage) algorithm (Application to Simulation of Chromosome Aberrations) [40] models human chromosome geometric configuration as well as simulates the radiation-induced breaks and their repair.

In clinical radiobiology, biophysical models must fulfill the following important criteria: to be mathematically simple, accurate, easy to handle and make use of a limited number of degrees of freedom in order to predict cell survival under irradiation. The so-called ‘Multiscale Approach (MSA)’ [41,42,43,44] offers a physical phenomenon-based analysis of the physical processes that take place on different spatial, temporal and energy scales in the micro-environment of the cell after its irradiation by ions [41]. The MSA includes many relevant effects known so far by the scientific community, which occur inside the cell when irradiated, starting from the physical stage, entering the chemical one and following the whole way down to the biological stage. This approach can be expanded to include secondary theoretical physical processes like nuclear fragmentation [45], charge-exchange processes through collaboration with other codes [46] and is characterized by adaptability to changes of external conditions, e.g., the presence of sensitizing nanoparticles [47]. It also comprises the contribution of new damage mechanisms, such as the thermo-mechanical effect with the formation and development of shock waves on the nanometer scale around the ion path [48] which might contribute drastically to DNA damage especially for ions with LET > 1000 keV/μm [49] (as those produced in accelerator laboratories or found in cosmic rays). Although such waves have not been detected experimentally so far but only described theoretically [50,51], their existence, as a result of the rapid increase in temperature and pressure in a close distance around the ion’s path, challenges radiobiological research to take action. The MSA has been used to predict survival probabilities of irradiated cells with ions of different charge and energy, and calculate RBE and oxygen enhancement ratio (OER). At this point it must be mentioned that the original MSA approach does not offer capability for survival prediction under a dose fractionation pattern in contrast to more established rather empirical models like LQ, MRM, LEM or others with a proven clinical use. 

In this paper, the methodology followed is described and applied to the analysis of measurements for ions studied or already used in hadron therapy, with the introduction of some crucial modifications to the original model. In this improved adaptation of ‘MSA model’, the complex DNA damage was assessed together with the obtaining of the survival curves of different cell series irradiated with heavy ions of different energy and LET. The OER and RBE were calculated and compared with experimental data. The ions studied cover a large spectrum of ion irradiation applications, not only in radiation therapy but also in radiation protection (especially ^56^Fe, which is considered as one of the most harmful components of cosmic radiation).

## 2. Results

In Section 4, we analyze the whole methodology followed as well as the parameters used in our model in order to obtain the final results. Therefore, having followed this methodology, we depict (Figure 1) the survival curves of different asynchronous cells for irradiation with some heavy ions with random values of LET and we compare them to the corresponding experimental ones from various references. In this model, we used the values—for the cell-specific input parameters—given by the corresponding references in which the experimental curves are demonstrated. As can be seen, for both values of the coefficient γ we obtain survival curves in very good agreement with the experimental data.

We now proceed to the calculation of the oxygen enhancement ratio (OER) at the 10% survival level for V79 (Chinese hamster) cells irradiated with ^12^C carbon ions. OER is defined as the ratio of the dose delivered to the cell under hypoxic conditions to that under normoxic conditions, leading to the same biological effect [52]. In our case:OER = *d*_10*,hypoxic*_/*d*_10*,normoxic*_(2)
where *d*_10,*normoxic*_ the dose at 10% survival level in normoxic conditions, and *d*_10,*hypoxic*_ the dose at 10% survival level in hypoxic conditions. We calculate the values of *d*_10_ through the survival curves for V79 cells for normoxic and hypoxic conditions (for γ = 0.0001) as obtained by our method, and then we calculate the OER; after that we compare these values of OER with the corresponding ones from [53].

Thus, we obtain the diagram of the OER as a function of LET (Figure 2). As can be seen, our results describe in a satisfactory way the decrease in the OER with LET increase and its asymptotical tendency towards the value 1 at high LET values. The values of the OER as obtained through our approach are in very good agreement with the experimental data. Under hypoxic conditions, when there is a reduction in oxygen concentration in the cell, there are less free radicals formed than under normal aerobic conditions, which leads to a decrease in the DNA damage caused by them.

Our final calculation is that of the RBE. This quantity for each cell line is defined as the ratio of the *d*_10_ to that of X-rays under aerobic conditions. All the data are the ratio to survival after irradiation with 200 kVp X-rays under normoxic conditions. We have used the survival curves as obtained via our methodology for some different cell series irradiated with ^12^C ions of different values of LET in the range 10–500 keV/μm. Our reference data are taken from [53,54]. In Figure 3 the diagram of the RBE at the 10% survival level is depicted as a function of LET for γ = 0.0001. As we notice, the RBE_10%_ increases with LET, reaches a peak at around 140 keV/μm (for V79 cell line, which has the biggest spectrum of LET values compared to the other corresponding ones) and then decreases at larger LET values. This feature constitutes the so called ‘overkill effect’ [61,62]; at high LET values the energy is deposited into the cell nucleus by a small number of ions. This energy exceeds the one required to kill the cell. One can notice that the calculated values of RBE_10%_ are in good agreement with the experimental results.

Although not shown, the OER and RBE values calculated with γ = 0.01 (as deduced in the present study) lead to almost identical results (within 10%) to those obtained for γ = 0.0001, thus, offering equally good agreement with the experimental data and other studies depicted in Figure 2 and Figure 3.

## 3. Discussion

Modeling the damages induced by such ionizing particles crossing the living tissues requires a full and accurate knowledge of the whole radiation history, the energy deposited during inelastic collisions and the part of energy which is transferred to secondary particles produced by them into living matter. Analyzing the scheme of the MSA, one should start with the propagation of an ion with a certain energy through biological matter, which is replaced in our calculations by water, since the largest percentage of a cell is of that [63]. Thus, liquid water is widely accepted as a tissue-equivalent medium for modeling the charged particle-induced processes in biological matter exposed to ionizing radiations.

The use of interaction cross sections for water medium—instead of DNA—is justified on the basis of the following arguments: The interaction cross sections of DNA are not as well-understood as those for water. For example, experimental energy-loss studies for DNA (either as a whole or for its individual bases) are scarce, while theoretical calculations are too uncertain due to its complex electronic structure (e.g., too many subshells, strong long-range correlation, influence of molecular geometry). Thus, it is not clear yet how accurate the existing calculations are using semi empirical dielectric response functions of solid-DNA which neglect the geometric structure and consider the energy distribution of the numerous subshells very roughly (see [64,65,66,67,68]). Similarly, it is not clear how realistic the atomic calculations are that pertain to the individual bases which neglect long-range correlation and polarization effects that strongly influence the most important outer-shell electronic transitions [69,70,71,72]. In general, the discrepancy in the inelastic and stopping cross sections among different DNA studies seems much larger than possible differences between DNA and liquid water, which seem to be of the order of 20–30% over the present energy range. In contrast, for water, there is a rich body of literature on theoretical calculations and experimental measurements for the interaction cross sections both in its gaseous and liquid phase, along with several intercomparison studies of different physics models [73,74,75,76]. However, many uncertainties still remain for water, especially in the low energy range [77,78,79], there exist several well-established physics models (see Emfietzoglou et al. [78]). Therefore, most Monte Carlo track structure codes for DNA damage calculations (e.g., PARTRAC, KURBUC, GEANT4-DNA, among others) still use interaction cross sections for liquid water to model the interaction of charged particles with DNA [71]. This approximation was also adopted here. 

Such an ion-projectile moving through matter interacts with a few bio-molecules along its path. The greatest damage to the cell is induced by the secondary electrons ejected as a result of excitation and ionization of the medium after the ion’s passage and the free radicals which are inevitably formed and propagate through the medium (direct damage). Most of these electrons have energies below 45–50 eV and undergo elastic and inelastic collisions with bio-molecules, but the main mechanism of bio-damage is the latter one. These secondary electrons share the following characteristics during propagation: their range of movement in bio-matter is up to 10 nm, their angular distribution of velocities after ejection is largely uniform, they lose most of their energy within only a few nm from the ion’s path and then continue to move until they become bound or until the formation of solvated ones; this occurs within 50 fs from the ion’s passage through the biological matter. Moreover, the elastic and inelastic scattering of these electrons prove to be nearly isotropic. It is of note that these secondary electrons have the potential to ionize just one or two water molecules and this could be explained by the fact that since the average energy of secondary electrons in the vicinity of the Bragg peak is about 40 eV (the ionization potential of water molecules I_w_ = 10.8 eV) the maximum average energy that can be transferred to a secondary electron of the next generation is ~(40 eV − I_w_)/2 = 15 eV, a value of energy which can barely cause another ionization [41]. All these features allow the use of the random walk approximation for the study of their transport.

The secondary electrons with energies more than 100 eV (δ-electrons) have been excluded from the random walk approximation; they are characterized by mean free paths of more than a few nm, and they are emitted mostly at small angles. Although they carry the potential to produce further ionizations than the former ones (those with energies lower than 50 eV), they do not spread much further than them. On the other hand, the probability of generating an electron with energy that exceeds 100 eV or even much more than that, is small compared with that of an electron which has a very lower energy. As a result, their inclusion in this model is ignored [41]. 

The physical description of an ion’s propagation through living tissues is obtained through a quantity called stopping power S_e_ (or equivalently, −d*E*/d*ζ*, i.e., the loss of ion’s kinetic energy *E* per unit path length *ζ*). In the case of ions, there is little difference between the position of their energy loss and that absorbed by biological matter along their path, since the ion’s energy is distributed to secondary electrons which have much shorter ranges in comparison with the length ζ. Therefore, to a first approximation the linear energy transfer (LET) can be considered identical to the stopping power. The main physical characteristics of the aforementioned secondary particles can be obtained through the calculation of the singly—differential cross sections (SDCS) (Equation (8)) for the ionization of water molecules, the total ionization cross sections (TICS) (Equation (11)) and their average ejected energy ($W_{average}$) (Equation (12)). The above mentioned physical characteristics (SDCS, TICS and $W_{average}$) of the ions that have been used in the methodology of the ‘MSA’ in this paper (protons, ^11^B, ^12^C, ^14^N, ^16^O, ^20^Ne, ^40^Ar, ^28^Si and ^56^Fe) are based on dielectric formalism [80] with the use of a semi-empirical parameterization [64,65] of the optical energy-loss function (OELF) (Equation (3)) of liquid water as target medium for this scope. 

Another assumption adopted in this model is that the scattering of secondary electrons has a cylindrical symmetry with respect to the longitudinal axis *ζ* of the ion’s path; this is because a typical range of the diffusion distance of a secondary electron in the Bragg peak region is less than 10 nm while the generating ion crosses a distance of ~1 μm. Simultaneously, the number of emitted secondary electrons per unit length ζ does not depend on *ζ* since the stopping power is approximately constant along a path of that range. 

Towards quantification of complex DNA damage in this approach, we calculated the average number densities of the secondary electrons produced on the ion’s path as a function of time and distance. Taking a step further, we obtained a corresponding number density of the new electrons induced by ionizations of the first generation of secondary electrons, since many of the latter have the potential to ionize molecules of the medium. After these second ionizations, the ionized molecules can become sources of reactive oxygen species (ROS), which play a basic role to lesions in the irradiated cell. Selecting the proper time fraction, it is feasible to obtain the initial spatial distribution of free radicals (especially hydroxyl radicals, OH^•^) (Equation (13)) which react with DNA and cause their own damage to this macromolecule. 

We also calculated the secondary electrons fluence $F_{e}\left(\rho \right)$ (see Equation (16)) (i.e., the integral of the number of secondary electrons that reach and impinge on a unit surface of the target per unit time, over the entire time after the ion’s crossing and over the target’s surface), which is dependent on the radial distance from the ion’s path *ρ* and the geometrical orientation of the surface-target. The calculation of this quantity is made for a fundamental element—target of DNA: two DNA twists, part of the macromolecule that is wrapped around the lateral surface of a nucleosome. This surface-target of DNA occupies 2.3 × 6.8 nm^2^ on the surface of a cylinder (histone octamer) (the radius of that cylinder is 5.75 nm) [63]. All the calculations have been made for the case where this cylinder is oriented in a perpendicular position with respect to the axis *ζ* of the ion’s path; the numerical results for this geometrical case are proved not to deviate significantly from other possible geometrical orientations-positions of the cylinder with respect to axis *ζ.*


Another basic physico-mathematical quantity which intercedes in the whole analysis is the probability of a secondary electron incident on a DNA twist to induce a single strand break (SSB)$\;\Gamma_{SSB}$; its values range in a wide interval, depending on the electron’s energy and the irradiated environment of a certain medium, according to different studies [81,82,83,84,85,86]. The chosen value for this probability used in our calculations (see Equation (17)) is that for the corresponding calculations for plasmid pBR322, as was estimated through fitting to experimental data and adopted by the authors of the MSA model [41]. This probability is important for the calculation of the number of SSBs induced per unit length of the ion’s trajectory, per ion.

The exact calculation of the number of DSBs induced per single cell can be characterized as elusive, since there are many indistinct views about the mechanism of causing such lesions and their induction. DSB is defined as two bistranded SSBs which occur on the opposite strands of DNA within a distance of ~10 bp [87]; in our calculations, within a single twist of DNA. DSBs are generally considered as serious lesions, still capable of being repaired by the existing endogenous repair mechanisms of the cell, but getting complicated and lethal for the cell’s life if other lesions occur close to them [88]. There are studies which suggest that a DSB may occur of a single electron that dissociatively attaches to a part of the DNA [9,89]. Another aspect about the production of a DSB is the one which inculpates two separate SSBs on opposite strands. In the latter case, double ionization events involving three electrons may cause such a lesion [90]. In this situation, a remark that, the ratio of the number of DSBs and SSBs per unit length of the ion’s path is constant and does not depend on the dose—only if tracks of different ions do not interact—gives us the solution to our question. The latter is proved by the fact that an increase in dose does not necessarily mean that there is also an increase in the density and interaction of ion tracks. Those tracks could overlap only when the dose has reached a critical value; this condition has not been observed in experiments so far [41]. Thus, the calculation of the DSB yield per single ion incorporates two different cases; the first where SSBs are converted into DSBs and the second where DSBs are induced by separate electrons. In our calculations, in the former case, the fraction λ of SSBs converted to DSBs (see Equations (18) and (23)) has taken the average value 0.15 (its values range between 0.1 and 0.2) [9,81,89]. The calculation of complex DNA damage in our model, following the methodology proposed by the authors of the Multiscale approach [44], implies that DSBs occur via SSB conversion since there is adopted a single-hit scenario, which means that a hit of an ion may inactivate a cell with a given probability.

Apart from SSBs, DSBs and other isolated base lesions which occur in the DNA molecule after cell irradiation, there are also clustered damage sites (multiply damaged sites) i.e., complex DNA damage, which are defined as a cluster of DNA lesions (SSBs, DSBs, damaged bases, etc.) within maximum two DNA twists. These ensembles of lesions which are treated as a single damage site by the cell’s repair mechanisms, determine the criterion of cell lethality. Precisely, since cellular repair as an individual process has not been included in the calculations of DNA damage of this model, the classification of DSBs on the basis of increasing complexity made by Schipler and Iliakis [88] has been adopted (through the parameter ν, in Equation (23)). It was assumed that a lesion which comprises a DSB and at least two other simple lesions within two DNA twists (T3-DSB, according to the aforementioned classification) is potentially lethal. The latter type of lesion is characterized by an increased complexity and in this case the cell energizes the simultaneous use of two repair pathways (e.g., DSB repair and BER, [60]). If we consider further a class of a superior order of complexity from this classification (e.g., T4-DSB, which represents a non-DSB damage cluster), the type of complexity gets higher, engaging potentially a different choice for the appropriate repair pathway that has to be followed. 

On the other hand, according to the relevant bibliography, there are obvious inconsistencies indicating gaps in our knowledge regarding the parameters which determine DSB repair pathway choice and the rationale on which this choice is based [91,92]. For example, we still cannot answer the question of why cells choose to follow an error prone repair pathway when an error-free repair pathway is available [93]. In such cases, things point to issues that need to be made clear when the network of repair pathways and the decisions that underlie their choice are analyzed [94,95]. Thus, we made use of an indirect way of characterizing whether a complex lesion is lethal or not: By introducing the criterion of lethality, we have tested successfully a large number of cell lines in different conditions of irradiation (different ions-projectiles with low to high LET values, small to large values of fluence, as well as aerobic and hypoxic environment) and we have demonstrated representatively some of them in our study. According to this criterion, the probability of lethal damage is described through Equation (23), where the order *ν* of complexity of a clustered damage site (i.e., the number of simple lesions in a cluster) is larger than or equal to 3. In this way, by introducing a weight coefficient corresponding to the probability of repair, we partially overcome the lack of a DNA repair component, as it is assumed that the clustered damage of the order of complexity equal to three or higher containing a DSB is lethal to the cell. It is natural to expect that sites with clustered damage of high orders are lethal for the cell, while the decreasing order increases the probability of a successful repair. We already know from a plethora of relevant studies [91,96,97] that lethal damage in DNA depends on the degree of damage complexity and dose. In general, our model includes, in an indirect way, the process of DNA repair which is embedded into this general criterion. Of course, this criterion may differ among different cell lines, but it expresses the average cell response to ion irradiation. This model, in its present form, is proved to simplify the way of assessing lethal damage to irradiated cells—in comparison to other radiobiological models—considering that inherent repair of DNA damage is included in the aforementioned criterion. 

Important to our assessments is also the average number of ions passing through the cross-sectional area of the irradiated cell nucleus *N*_ion_ (Equation (25)) as well as the average distance of the ion’s crossing through the nucleus of the cell $\bar{X_{nc}}$ (Equation (28)). Finally, an equally critical quantity for the calculations of clustered DNA damages in irradiated cells is the number of clustered damage sites per unit length of the ion’s path$\;n_{s}$, (Equation (20)) which depends on the genome size of the cell, the nuclear volume and the phase of the cell cycle. In the latter case, the number density of DNA is averaged over the different phases of the cell cycle in order to give a certain value that fits to an average cell.

A very important calculation for the assessment of complex DNA damage in the MSA, and generally for obtaining the survival curves, is the average number of multiple damage sites per cell induced by free radicals; more precisely, by hydroxyl radicals OH^•^. Because among the reactive species produced by water radiolysis, the charged species H^+^ and OH^−^ are quickly neutralized (thus, they are not considered harmful). Similarly, solvated electrons and H atoms do not induce DNA strand breaks since the electrons add to the DNA bases (this fact does not induce a breakage of a phosphoric acid ester bond of the sugar moiety). Therefore, OH^•^ radicals add primarily to the double bonds of the bases and abstract H atoms from the sugar moiety to a large extend. These radicals (OH^•^) are mainly responsible for the induction of strand breaks in a DNA molecule [98]. As a whole, for the estimation of the average number of multiply damaged sites per cell induced by OH^•^, an equation is applied (Equation (21)); this equation comprises a nearly constant average value of area density of hydroxyl radicals—calculated indirectly from the average value of the number density of hydroxyl radicals within a distance of 10 nm from the ion’s path—and the Heaviside step function which describes accurately a considered uniform distribution of these species within that distance from the ion’s trajectory. In this case, the presence of hydroxyl radicals beyond the distance of 10 nm from the ion’s path is totally excluded. This product is multiplied by the probability $\Gamma_{r,SSB}$ of any OH^•^ reaching the DNA to produce a SSB. 

Last but not least is the development of a factor—part of the Equation (16) which calculates the fluence of the secondary electrons through our target; that is the attenuation exponential factor e^−γ*k*^ which discerns electrons that do not take part in the random walk. The coefficient γ is defined as the ratio of the ionization cross sections of secondary electrons with a minimum energy capable to induce DNA damages to the total cross section (The total cross section is the sum of the excitation, ionization and elastic cross sections, respectively) [99]. In our paper, we examined two different values for the coefficient γ: the first one (γ = 0.0001), adopted by the authors [100], corresponds to secondary electrons with energy equal to or higher than the ionization potential of water molecules (10.79 eV) [101], while the second one (γ = 0.01) is an alternative value for the case that the energy of these electrons is equal or larger than 17.5 eV (this is the threshold energy for the induction of an SSB, according to [87]). In the latter case, the probability of an SSB induction $\Gamma_{SSB}$ is considered equal to 1.

We showed that our MRM model can efficiently predict the survival curves of different human cells irradiated by ions in very good agreement with the experimental data without using any kind of fitting to experimental data, by utilizing only three input parameters (particle’s LET or kinetic energy *T*, and two cell-specific parameters: the cross sectional area of each cell nucleus Α_n_ and its genome size N_g_). In fact, when we study human cells, these cell-specific parameters may be reduced to only one; the cross sectional area (another parameter used in our model is $N_{rk}$ which may take any value ≤15 × 10^−3^ nm^−2^, thus it is a default one). We also showed through this model that one may safely use another alternative way to that proposed by the authors of the ‘MSA model’, of considering the damage caused to DNA by the secondary electrons, simply by entering a different threshold of their energy when hitting the target (through the coefficient γ). It is of note that our results are independent of the cell cycle phase in which the cells are irradiated since this model incorporates all the phases of the cell cycle by introducing an average number density of DNA over the different phases as a function of fractions of the total cell cycle duration [49].

In this study, we have used the parameter N_g_ as the genome size (the number of base pairs in the cell nucleus) considering that in the interphase most of the chromatin is relatively decondensed and uniformly distributed throughout the nucleus. This means that the corresponding yields for all normal human cells will be the same; of course, this does not happen in *cancer cells* which are characterized by an elevated mitotic index [102]. In this model, and in the case of cancer cells which are generally characterized by unregulated growth, and larger nuclei [103], by giving the corresponding measured values of the cell parameters N_g_ and A_n_ for every studied cell, we efficiently predict cell survival after their irradiation with heavy ions. In this way, we prove the model’s adaptability tο any alternation between normal and cancer cell lines, and vice versa. For example, through our analysis, the survival curves of the normal human skin fibroblast (NB1RGB) cells (number of chromosomes: 46.3 and area of cell nucleus: 172.3 μm^2^) and the brain glioblastoma (KS-1) cancer cells (number of chromosomes: 83.3 and area of cell nucleus: 221.4 μm^2^) [54] after their irradiation with different ions and different LET values, as depicted in Figure 1i,iv,v, show a very good agreement with the experimental data. We must note that in our present results, the survival curves depicted in Figure 1 are representatively part of a large number of cell lines studied by us through the present model, and a significant part of experiments that have been included and summarized in the PIDE database [104].

As can be seen from the survival curves presented in our study, the performance of our model at low-medium doses is satisfying enough, since all the curves show a very good agreement with the experimental data at doses lower than 2 Gy (2 Gy/day is a typical dose fraction in radiation oncotherapy) except for that of Figure 1vi, for V79 cells irradiated with ^56^Fe with LET = 2106 keV/µm (an ion studied especially in cosmic radiation field, and not in radiation therapy) because of the absence of experimental data in this low LET area).

We may now discuss the limitations of the present model. The sensitivity of the MRM model to parameters changes is proved to be very high. Actually, except for the coefficient γ -which is analyzed in Section 4—and, when changed, this happens simultaneously with a change in the probability $\Gamma_{SSB}$ of a SSB induction, the other non-variable parameters are: (i) the probabilities $\Gamma_{SSB}\;$ and $\Gamma_{r,SSB}\;$for the induction of a SSB on our elemental DNA target due to an electron, or an OH• incident on it, respectively, (ii) the conversion probability *λ* of a SSB to a DSB and (iii) the order *ν* of damage complexity (in Equation (18)). More analytically: (i) The ‘famous’ value 0.13 for the probability $\Gamma_{r,SSB}\;$is proved to be constant in literature over the years [87,105,106], and we have also kept it like this in our study. If we use any other value between 0.14 and 0.22, according to Van Rijn et al. (1985) [107], our final results are not affected at all. On the other hand, this also does not happen with the probability$\;\Gamma_{SSB}$, which has been chosen by us taking the value 0.03, according to the authors of the MSA [41], but may vary from 10^−4^ to 10^−1^ [81,108]. In our calculations, any value of that probability from 10^−4^ to 8 × 10^−^^2^ does not affect the value of the number of lethal DNA lesions per ion *Y_C_*, or the slope of the survival curve for any cell studied. For values larger than 8 × 10^−2^ there occur extensive uncontrollable declinations in our results from the experimental data. Besides, for values lower than 10^−4^, our final results do not change at all. (ii) About the parameter λ, we have already analyzed its values given by us above. (iii) As for the parameter *ν* which also has been discussed above, we may note that if we tried to use other values lower or higher than the selected one (*ν* = 3), then this would mean that we either take into account lesions which are easily repaired and are not considered lethal (*ν* = 1 or 2) [88], or (for *ν* = 4) we exclude from our calculations lesions (i.e., T3-DSB) which are difficult to be repaired and they are considered lethal; in this case, our calculations lead to completely wrong results.

Radiotherapy (RT), besides cancer cells, affects also the tumor microenvironment (tumor stroma); the latter surrounds the cancer cells and it consists of non-malignant cells, extracellular matrix, as well as cells of the immune system [109]. Tumor microenvironment’s complexity is strongly connected to tumor growth, metastasis, response to therapy and radioresistance [110]; the latter remains until nowadays the main reason for the failure of treatments at cancer patients. Until recently, RT was focused only on the direct tumor cytotoxicity and the induced DNA damage. However, since the immunomodulatory effects of ionizing radiation on the tumor microenvironment have been steadily recognized, there is a strong interest of the scientific community in utilizing radiotherapy to activate an anti-tumor immune response [111]. RT participates in releasing and presenting the antigens, gathering T-cells in the tumor, activating T-lymphocytes and stimulating them to recognize and kill the cancer cells [112]. During all the tumor’s life, a mutual communication takes place between the cancer cells and their surrounding tumor microenvironment’s components; they dynamically interact with each other, controlling initiation, progression, invasion and metastasis of the tumor [113]. Generally, there is proved to be a delicate balance between the immune system activation and the immunosuppression as they are both induced by radiotherapy; and this balance depends on the fractionation scheme and dose [114]. Up until now, according to the bibliography reports, there is no clear evidence about which doses stimulate the activation of the antitumor immune response and which of them provoke immunosuppression [115]. In any case, our model may contribute to the therapy planning, since its results are proved to be satisfactory even in the very low doses.

Since one of our model’s goals is the optimization of ion-beam therapy, we must also keep in mind the principle of fractionation of the radiation dose; this may be synopsized as follows: by dividing a dose into several small fractions, normal tissues are spared due to repair of sublethal damage between dose fractions and repopulation of the cells if the whole time is long enough. Simultaneously, damage to tumor cells is increased because of reoxygenation and redistribution of the irradiated cells into radiosensitive phases of the cycle between dose fractions. This technique produces in general better tumor control for a certain level of toxicity of normal tissue than a single undivided dose [116]. Because our model functions very efficiently in the ‘low’ dose area, it can potentially be used in the future for predicting the dose-response relationship for a multifraction regimen. Currently, this is a limitation for MRM compared to other more advanced but rather empirical approaches as summarized in [117].

Up until now, we have not considered DNA repair; it is assumed that inherent repair of DNA damage is included in the criterion of lethality, as was analyzed above. We may say that this criterion together with Equation (29) produce linear survival curves for ion irradiation of cells. Nevertheless, the existence of ‘shouldered’ survival curves also in experiments for specific cells could be partially translated—through the methodology and the principles of the MRM—as the fact that there are still potentially ‘lethal’ damages which can be repaired by the irradiated cell. It is of note that an extended research on cells irradiated with X-rays and their survival curves [52] shows that the linear survival curves indicate a single-hit scenario of DNA damage, which also means that a single hit of a cell’s DNA-target with, in our model, an ion leads to inactivation of that cell with a given probability; a probability which includes the probability of DNA repair. On the other hand, linear survival curves are a signature of cells exposed to densely ionizing radiations (such as heavy ions, used in radiation therapy of cancer [58]). 

Considering possible future developments of the MRM, we could focus, at first, on the cellular repair mechanisms which reduce the number of complex lesions through processing after subjection to ionizing radiation. Currently, we know that different pathways have been evolved to amend DSBs or generally complex damage: the homologous recombination (HR) repair pathway, the DNA-PK- dependent non-homologous end-joining (D-NHEJ) and the (back-up) B-NHEJ [118]. Modeling approaches based on biochemical kinetic equations which would comprise simultaneously all these different repair pathways would be welcomed to this model. 

The original ‘MSA’ also incorporates the theory of shock waves generated by the developed high pressure inside the cylindrical surface of radius ~1 nm that encircles the ion’s path. Although such waves have not yet been discovered experimentally, their existence is expected according to thermodynamics. The action of shock waves which are predicted by this approach may be critical for the DNA damage, especially for that percentage of damage induced by free radicals. This is based on the sudden expansion of the medium which may transfer these radicals further and more effectively than the classical mechanism of diffusion. However, in this work, we did not take into account this potential mechanism, its study and thermo-mechanical analysis maybe of importance for assessing more accurately DNA damage for future developments. 

Up until now the radiobiological models have been focused on the damage induced to the DNA of an irradiated cell, but new indications from recent research results show that this may not be the only leading target of radiation. Breakage of cellular DNA after radiation occurs in both the nuclear and the extra-nuclear DNA. Besides nuclear (nDNA), mitochondrial DNA (mtDNA) is also affected by IR; the latter is proved to be more susceptible to IR than the former one [119]. Many investigations showed that mtDNA can be an easily available target for free radicals [120,121]. These organelles may occupy up to 30% of the total cell volume [122]. IR can induce various lesions in mtDNA (such as strand breaks, base mismatches and large deletions [123]. A large number of studies show that mitochondria and their interactions with the nucleus play a basic role in the induction of oxidative stress post irradiation [124], in the epigenetic changes as well as in genomic instability [125]. They also initiate and amplify bystander signals [126]. Thus, a possible future extension of our model to assess potentially lethal lesions in the mtDNA—concentratively in an irradiated cell—could contribute to an increase in the tumoricidal efficacy of a targeted radiation therapy. 

Another issue for embedding in this model is that of bystander effects after cell irradiation, which means the occurrence of biological effects in unirradiated cells as a result of exposure of other neighboring cells of the same population to ionizing radiation which show a non-linear response to dose [127,128]. Cells that are not directly hit by ions (often called bystander cells) may exhibit responses similar to irradiated cells. Cell culture experiments and in vivo observations have shown the induction of DNA DSBs, mutation, chromosomal aberration, apoptosis, and genomic instability [129]. It is of note that bystander effects are characterized as a low-dose phenomenon [101]. Since our presented model has the ability to predict the survival fraction of irradiated cells with ions of different LET, this fact in combination with experimental data from irradiated cell lines could give a more accurate prediction of survival curves even for surrounding non-hit cells [130]. 

In the same way, we may refer to the radiation adaptive response, a phenomenon that was first reported by Olivieri et al. [131] and followed by other studies [132,133]. According to this phenomenon, human lymphocytes can become ‘adapted’ by prior exposure to low level irradiation (‘adaptive dose’) so that they become less sensitive to the chromosome-breaking damage due to high-dose X-rays (‘challenging dose’) delivered subsequently. Therefore, this adaptive response to low doses occurs only within a relatively small ‘window’ of dose [134]. This means that very low doses of IR in human and animal cells could induce mechanisms whereby cells become better adapted to confront subsequent exposures to high doses [135]. The adaptation induced by low doses of IR is attributed to the induction of a repair mechanism; if the latter is active at the time of exposure to high doses this would lead to less residual damage [134].

Eukaryotic cells are proved to have radiation-inducible DNA repair mechanisms that may be regulated in response to DNA damage. This fact has a practical importance especially for clinical application in radiation oncotherapy when designing the proper dose fractionation, thus, it is connected to the split dose protocols and the analysis of ‘irregularities’ in dose-response curves; this also means that it is relevant to the analysis of effects of very low doses or low dose rates in human cells [136,137]. There are also findings suggesting that bystander effects induced by irradiation of cells might contribute to the induction of the radiation adaptive response [138]. These phenomena are of great importance at low doses and have an impact on the shape of the dose–response relationship. The high sensitivity of our model in low doses, as has been mentioned above, may contribute to the observation of these phenomena, too.

Thus, there are many challenging research areas in which the MRM model could be expanded in the near future, with the accumulation of more experimental data. Its versatility makes it adaptable to modern experimental conditions and results since it is based on a complete physical, biological and chemical framework.

## 4. Materials and Methods

In this work, in all of our equations, we have kept the same symbols for all the quantities described by the authors of the ‘MSA model’, in order to facilitate the readers when we cite their references. All our results were extracted with the use of the MATLAB software (R2019a, MathWorks). Before proceeding to our calculations we present a flow diagram of the whole methodology of our model (Figure 4). 

The calculations of singly-differential cross sections (SDCS) (i.e., the total ionization cross sections of water molecules differentiated with respect to secondary electron kinetic energy) are based on the dielectric formalism [139] with the use of the energy-loss function (ELF) Im[−1ϵ(k,E)] calculated for liquid water, where ϵ(k,E) is the complex dielectric function, and E and *ħ × k* are the energy and momentum transfers, respectively. Since ELF is experimentally known, one can compute the ionization cross sections. This function can be determined experimentally from optical data [73,140]. 

In order to calculate ELF, one has to calculate at first the optical ELF (OELF) for zero momentum transfer *k* = 0. OELF may be calculated from experimental optical data for a large number of materials, of which biological ones are only a few [66]. For its calculation, we have used an empirical approximation also suggested by [141] which is based on the observation that some bioorganic compounds along with water have a rather similar absorption spectrum at low energy transfers. These authors have parameterized the experimental OELF with a single-Drude function:(3)OELF=Im[−1ϵ(k=0,E)]=α×Ε[E2−Εp2(Ƶ)]2+γ2(Ƶ)Ε2
where α, *E_p_*(Ƶ) and *γ*(Ƶ) are the intensity, position and width of the single-Drude ELF, respectively, and Ƶ is the mean atomic number of the target. The above expression is a simplification of more elaborate versions which aim to account for all the different energy-loss channels [74,142]. Following this method and giving the fitted parameter values suggested in that paper for liquid water (i.e., Ƶ = 3.333, α = 3856.3 eV^3^, *E_p_*(Ƶ) = 23.192 eV, *γ*(Ƶ) = 14.811 eV), the calculation of ELF in the optical limit may be then extended to *k* ≠ 0, and thus, lead one to obtain ELF. Using the dispersion relation introduced by Ritchie and Howie [143] together with the following equation [64] for *E_p_* (*k* ≠ 0):(4)$$E(k\neq \;0)\;=\;{Ε}_{p}\left(k=0\right)\;+\;ꬰ\;\times \;(\hbar^{2}k^{2})/2\;m_{e}$$
(for liquid water: ꬰ ~ 1, [64]), where ${Ε}_{p}\left(k=0\right)=\;$*E_p_* (Ƶ) and m_e_ is the electron mass.

Thus, ELF (*γ*(Ƶ) = *γ*):(5)ELF=Im[−1ϵ(k,E)]=α×Ε[Ε2−Εp2(k)]2+γ2Ε2

More elaborate dispersion relations have been developed for liquid water [78,144] and can be used in future extensions of the MSA.

The energy transfer is:(6)$$E\;=\;W\;+\;B_{i}$$
where *B_i_* is the binding energy of the i-shell. For water, we keep the estimate of de Vera et al. [64] of a mean binding energy *B* from the ionization thresholds of all outer electronic shells, and therefore: (7)E = W + B

(The mean binding energy for liquid water *B* = 18.13 eV). 

For liquid water, the microscopic singly-differential cross section (SDCS) $\frac{d\sigma }{dW\;}\;$[64] for an electron emitted with kinetic energy *W* from the electronic i-shell of the target, by an ion of kinetic energy *T*, mass *M* and charge Z is:(8)$$\frac{d\sigma }{dW}=\frac{e^{2}}{N}\times \frac{\left(M\right)\times Zeff^{2}}{\pi T\times \hbar^{2}}\times \int_{k^{-}}^{k^{+}}\left(ELF\right)\frac{1}{k}dk$$
with integration limits:$$\;k^{+}=\;\sqrt{M}\times \left(\sqrt{T}+\sqrt{T-E}\right)$$
$$k^{-}=\sqrt{M}\times \left(\sqrt{T}-\sqrt{T-E}\right)$$

The molecular density of water is: *Ν* = 0.033 molecule/Å^3^.

While SDCS depend mainly on the charge and the velocity of the ion that enters matter, and since charge is dependent on velocity, the formula for the effective charge Z_eff_ of an ion (with charge z) [145] has been included in our calculations:Z_eff_ = z [1 − exp (−125 β × z^−2/3^)](9)

Noteworthy is the fact that the macroscopic cross section Λ is related to the microscopic one, σ [64]:(10)Λ = N×σ

Now, one can proceed to the calculation of the total ionization cross sections (TICS):(11)$$\mathrm{TICS}=\Lambda \left(T\right)=\int_{0}^{\infty }\frac{d\Lambda \left(T,W\right)}{dW}dW$$
and the average energy of the emitted electrons: (12)$$W_{average}\left(T\right)=\frac{1}{\Lambda \left(T\right)}\int_{0}^{\infty }W\frac{d\Lambda \left(T,W\right)}{dW}dW$$

By the described method we processed the physical data of different ions, i.e., protons, ^11^B, ^12^C, ^14^N, ^16^O, ^20^Ne, ^28^Si, ^40^Ar and ^56^Fe, as projectiles in liquid water and we show our results in the following two diagrams (see also Appendix A). In Figure 5, the TICS of the studied ions calculated through this method are shown as a function of the ion’s kinetic energy T. From this graph one can notice that the TICS for every ion get their maximum in a certain different value of energy and these peaks shift to larger energies as the mass of the ions get increased. At the same time, TICS represent the number $\frac{dNe}{d\zeta }$ of secondary electrons emitted per unit length from the ion’s path, and from this graph is shown that the heavier the ion, the larger the number of ejected electrons becomes while it crosses biological matter. This number of ejected electrons contributes greatly to the damage in a cell, and especially in its DNA.

In Figure 6**,** the average energies$\;W_{average\;}$of secondary electrons produced are depicted as the result of ionization after the passage of these ions through liquid water as a function of the ion’s energy. As can be seen, the peak of the curve for every ion shifts to the right (to larger kinetic energies of the ion) as the mass of the ion grows larger but they all tend to a stable maximum value below 45 eV. This also means that the difference in the impact on living matter from its crossing by different ions will be focused on the number of secondary electrons produced per unit length of the ion’s path, and of course, on the corresponding number of OH^•^. 

The next step is the calculation of the OH^•^ number density in the medium after the ion’s crossing through it. For this reason, we follow the differential equations described by the authors [49], calculating the number density of OH^•^ for carbon ions ^12^C of a certain LET. We then extend it to every value of LET and to every ion-projectile. For this reason we use the mean free path data from [146,147]) taking into account that the average energy of secondary electrons ejected from all the heavy ions that we study is nearly the same, as referred above. Thus, we demonstrate the final form of the equations by which the initial number density of hydroxyl radicals $\;n_{ΟΗ•}\left(\rho ,t\right)$ (as a function of distance from the ion’s path ρ and time t) is acquired:(13)$$n_{ΟΗ•}\left(\rho ,t\right)=\;\int_{0}^{t}\frac{dN_{e}}{d\zeta }\times \delta^{2}\left(\rho \right)\times \delta \left(t\right)dt+\;\int_{0}^{t}\frac{n_{1}\left(\rho ,t\right)}{\tau_{1}}dt+\int_{0}^{t}\frac{n_{2}\left(\rho ,t\right)}{\tau_{2}}dt\;\;\;$$
where *δ* is the Dirac delta function, and the number density of first-generation secondary electrons $\;n_{1}\left(\rho ,t\right)$:(14)$$n_{1}\left(\rho ,t\right)=\frac{dNe}{d\zeta }\times \frac{1}{4\pi D_{1}t}\times e^{-\frac{\rho^{2}}{4D_{1}t}-\frac{t}{\tau_{1}}}$$
and $\frac{dNe}{d\zeta }$ the number of emitted secondary electrons per unit length (i.e., the value of TICS, as mentioned above), as well as the number density of second-generation secondary electrons $\;n_{2}\left(\rho ,t\right):$
(15)$$n_{2}\left(\rho ,t\right)=\frac{1}{2\pi \tau_{1}}\times \frac{dNe}{d\zeta }\times \int_{0}^{t}\frac{1}{D_{1}\cdot t^{\prime }+D_{2}\left(t-t^{\prime }\right)}\times e^{-\frac{\rho^{2}}{4\left[D_{1}\cdot t^{\prime }+D_{2}\cdot \left(t-t^{\prime }\right)\right]}-\frac{t-t^{\prime }}{\tau_{2}}-\frac{t^{\prime }}{\tau_{1}}}dt^{’\;\;\;}$$
while the diffusion coefficients *D_i_* = *v_i_*‧*l_i_/*6, where *v_i_* is the electron’s velocity and *l_i_* its elastic mean free path in the medium, with the values *D*_1_ = 0.265 nm^2^/fs, *D*_2_ = 0.057 nm^2^/fs and the average lifetimes of the two generations’ electrons, respectively, *τ*_1_ = 0.64 fs and *τ*_2_ = 15.3 fs (mean free path data have been taken from [146,147]). We have considered time t≃50 fs as that of integration over these equations, since at this fraction of time the transport of these electrons is over [49,130]. This time is shorter than that of the formation of hydroxyl radicals, but by the time OH^•^ are formed there are no more active sources. These radicals react with other biomolecules in a cell. Therefore, we demonstrate the graph (Figure 7) which shows the initial distribution of hydroxyl radicals for different heavy ions and different random values of the stopping power S_e_.

These radicals diffuse around the ion’s track at distances ρ ~ 6 nm, but not more than 10 nm [148,149]. This is why we studied the spatial distribution of them over a distance *ρ*
*=* 0–10 nm from the ion’s path. For all these different values of stopping power and all the ions that we study, we calculate the average value of the OH^•^ number density within this distance, which represents the number density of these radicals at a random point in a distance *ρ* from the ion’s path (where our target could be); its value proves to have an order of magnitude 10^−2^ nm^−3^. This approximation will be used in this paper for the assessment of DNA damage from hydroxyl radicals, and this is also an innovative method which deviates from that used by the authors of the MSA up until now.

### Calculation of Complex DNA Damage and Survival in an Irradiated Cell

Since we know that TICS expresses the number $\frac{dN_{e}}{d\zeta }$ of ejected secondary electrons per unit length of an ion’s path through biomatter, we calculate their values as a function of the ion’s energy *T* and stopping power S_e_ for all the ions that we study (see Appendix A). This will be important in order to calculate the fluence of secondary electrons through the selected target and, consequently, to assess DNA complex damage. We have used the data for protons and carbon ions in liquid water from the appendix of ICRU Report 90 [150], and the corresponding data for the rest of the heavy ions (except for _56_Fe) from the appendix of ICRU Report 73 [151], while for ^56^Fe the data have been taken from SRIM 2008 [152] (see Appendix A). In this way, one may address to these datasets and obtain the number of emitted electrons per unit length of the ion’s trajectory $\frac{dN_{e}}{d\zeta }$ for any value of the ion’s kinetic energy *T* or stopping power S_e_ in liquid water. 

Since we know the number of ejected secondary electrons from the ion’s path per unit length, we may calculate in a pure geometrical way the fluence of the secondary electrons through an elemental structure of DNA which has been chosen for target and this is the surface of two twists of DNA (part of the segment of DNA that is wrapped around the lateral surface of a nucleosome). Its dimensions are 2.3 × 6.8 nm^2^. Having mentioned previously the whole geometry of this target hit by secondary electrons and following the formulae described by [41] we give the final formula for this fluence $F_{e}\left(\rho \right)$ (for γ = 0.0001):(16)$$\begin{array}{c} F_{e}\left(\rho \right)=\;\frac{dN_{e}}{d\zeta }\times 5.75\times \\ \times [\int_{-1.15}^{1.15}dz\int_{-\infty }^{\infty }d\zeta \int_{A}^{B}d\phi \int_{\frac{r}{0.2}}^{\infty }\times \frac{r}{2k}\left(\frac{3}{2\pi k\times 0.04}{)}^{1.5}‧e^{\frac{-3r^{2}}{2k\times 0.04}-\;0.0001\times k}\times \frac{\left(5.75-\rho cos\phi -\zeta sin\phi \right)}{r}dk\right] \end{array}$$
with limits
$$Α=max\left[-0.591,\;\left(\arctan\frac{\zeta }{\rho }-\arccos\frac{5.75}{\sqrt{\zeta^{2}+\rho^{2}}}\right)\right]$$
and
$$Β=min\left[0.591,\;\left(\arctan\frac{\zeta }{\rho }+\arccos\frac{5.75}{\sqrt{\zeta^{2}+\rho^{2}}}\right)\right]$$
where
$$r=\sqrt{{\left(5.75\times \cos\phi -\rho \right)}^{2}+{\left(5.75\times \sin\phi -\zeta \right)}^{2}+z^{2}}$$

For the elastic mean free path *l* of secondary electrons with energies less than 45 eV we have chosen the value 0.2 nm (its values range between 0.1 and 0.45 nm, [146]. It is of note that the coordinate ζ is considered to range from −∞ to +∞, since the length scale along the Bragg peak is measured in tens of μm, while the radial one in nm. As mentioned above, we have used both the cases which discern the energy threshold of secondary electrons which hit our surface-target, as it is expressed through the coefficient γ. The used values of γ are calculated as the ratio of the ionization cross sections of those secondary electrons, with a certain energy threshold to induce damages to the target DNA, to the total cross section. The values of cross sections have been taken from [75]. Our final results about complex DNA damage printed in this paper, have been calculated by using both the values of γ and at last, these results are compared between them. 

We may now introduce the probability $\Gamma_{SSB\;}$of an electron incident on this target to induce a SSB. This probability can be estimated in several ways [81,82,86]. We have chosen the value $\Gamma_{SSB}=0.03$, since it is validated via experiments on plasmids by the authors [41] and used to great extent by them. Multiplying this probability by the fluence of secondary electrons through this target we get the average number of SSBs for this fundamental target-part of DNA:(17)$$N_{e}\;=\;\Gamma_{SSB}\times \;F_{e}\left(\rho \right)$$

Taking into account, until now, only the action of secondary electrons, we may express the probability $P_{e}\left(\rho \right)$ that the order *ν* of damage complexity at that given segment of DNA is equal or larger than 3, following the criterion of cell lethality (which is based on the classification of Schipler and Iliakis [88] which mentions that any complex DNA damage that contains a DSB together with at least two more simple lesions is potentially lethal for the cell, as it was described above:(18)$$P_{e}\left(\rho \right)=\lambda \times \overset{\infty }{\underset{\nu =3}{\sum }}\frac{N_{e}^{\nu }}{\nu !}e^{-N_{e}}$$
thus, we assume that such a clustered damage site of complexity order equal to three or more is lethal.

We also use λ = 0.15 [9] as the conversion probability of an SSB to a DSB, since in the MSA, DSBs take place only through an SSB conversion [89]. More simply, if a single electron induces a SSB, the same electron induces also a DSB with a probability 0.1–0.2 of that to cause an SSB.

The number of complex damage sites per unit length of the ion’s path $\frac{dN_{cd,e}}{d\zeta }$ through the cell due to secondary electrons is:(19)$$\frac{dN_{cd,e}}{d\zeta }=\int_{0}^{\infty }n_{s}\times P_{e}\left(\rho \right)2\pi \rho d\rho$$
where $\;n_{s}$ is the number of complex damage sites per unit length of the ion’s path (number density), which is proportional to the ratio of base pairs of the cell’s DNA to the nuclear volume, $n_{s}$∼N_bp_/V_n_. Here, we must take into account that a double DNA twist includes 20 bp [153] and the fact that in order to calculate the final expression of $n_{s}$ one must consider the dependence of N_bp_ on the phase of the cell’s cycle and then average it over these different phases. The final expression for $n_{s}\;$ [42]:(20)$$n_{s}\;=\;(\pi /16)\;N_{g}/({Α}_{n}\times \bar{{\;X}_{\mathrm{nc}}})$$
where N_g_ is the genome size (for human cells, it is equal to 3.2 Gbp [63]), Α_n_ is the cross sectional area of each cell nucleus and $\;\bar{X_{nc}}$ is the average length of ions’ traverse through nucleus.

The next step is to include the indirect effect of hydroxyl radicals, which induce in their own way damage to DNA. However, in this case, a corresponding fluence of hydroxyl radicals through the same target is not able to be calculated. However, we know the radial distribution of the hydroxyl radicals number density and what we seek is the surface density of them at any distance ρ in a cylindrical volume of radius 10 nm around the ion’s trajectory; for this reason, since we have no evidence about this magnitude, we consider values of the hydroxyl radicals surface density less by at least one order of magnitude than the average number density that we assessed above (i.e., values ≤ 10^−3^). These values must then be reduced to the surface of this elemental target of DNA, (i.e., 15.64 nm^2^); these are the values of $N_{rk}$ that we use below. Therefore, in this paper—deviating from the classical MSA—we introduce at first the step function, in order to describe the surface density of OH^•^ as a function of the distance ρ, and then we multiply it by the OH^•^ activation probability $\Gamma_{r,SSB}$, so as to calculate the average number of lesions like SSBs, base damages, etc., due to hydroxyl radicals$\;N_{r}$:(21)$$N_{r\;}=\;N_{rk}\times \theta \left(10-\rho \right)\;\cdot \;\Gamma_{r,SSB}$$
where $N_{rk}$ ≤ 15×10^−3^ nm^−2^. It turns out that for any value of $N_{rk}$ equal or less than this approximate value, we get results—about the surviving fraction of the cells—which are in very good agreement with the experimental ones, as will be shown below. The step function excludes radicals at distances larger than 10 nm from the ion’s path.

On the other hand, we have used the value $\Gamma_{r,SSB}$ = 0.13 for the probability of any OH^•^ reaching the DNA to produce a SSB [87,105]. This value is also estimated by Milligan et al. [106]. Other assessments for the value of$\;\Gamma_{r,SSB}$ range from 0.14 to 0.22 [107]. It was also supposed that hydrated electrons and H atoms react mainly with the bases of DNA [154] to induce base radicals, and thus, are excluded from our calculations. All the aforementioned refer to the classical normoxic conditions of the irradiated cell. In the case of hypoxic conditions, where oxygen concentration is very low or nearly absent, the DNA damage induced by radicals may be repaired and therefore we may consider that the average number of SSBs induced by radicals,${\;N}_{r}$ is about the half of that for normoxic conditions [5,155]. Thus, in our approach, only this parameter, $N_{r}\;$depends on oxygen concentration. In general, it has been shown that direct damage is important and for example constitutes ~70% or a total damage to cells by particles of 2 MeV/um, equivalent to ^56^Fe [57]. 

We then add the average number of simple lesions due to secondary electrons $\;N_{e\;}$ to that one due to OH^•^$\;N_{r}$, to find the total average number of such lesions $\;N_{c}$:(22)$$N_{c\;}\;=\;N_{e}\;\;+\;N_{r}$$

According to the criterion of lethality that has been set in the MSA, we may proceed to calculate the probability of lethal damage $P_{l}\left(\rho \right)$:(23)$$P_{l}\left(\rho \right)=\lambda \times \overset{\infty }{\underset{\nu =3}{\sum }}\frac{N_{c}{}^{\nu }}{\nu !}e^{-N_{c}}$$
where ν is the number of simple lesions per cluster.

The yield of lethal damages per unit length of the ion’s trajectory $\frac{dN_{l}}{d\zeta }$ is then:(24)$$\frac{dN_{l}}{d\zeta }=n_{s\;}\cdot \left[\int_{0}^{10}P_{l}\left(\rho \right)\times 2\pi \rho d\rho \;+\;\int_{10}^{\infty }P_{e}\left(\rho \right)\times 2\pi \rho d\rho \right]$$
since hydroxyl radicals act only within the distance *ρ =* 0–10 nm from the ion’s path in contrast to secondary electrons that are considered everywhere in the volume that we examine.

On the other hand, for a given type of cell irradiated with a given dose *d* (measured in Gy) of an ion beam (assuming that there is a uniform distribution of ions in the beam) with LET (or stopping power S_e_—in keV/μm—since these terms are considered synonymous in this approach), the average number of ions that cross the cell nucleus *N*_ion_ will be:*N*_ion_ = Α_n_ × *d/*S_e_(25)
and reducing it to the proper units:*N_ion_* = [Α_n_·10^6^ (nm^3^) × 10^−24^ × *d* (Gy)]/[S_e_ (eV/nm) × (1.6 × 10^−19^)](26)
where Α_n_ is the cross-sectional area of each cell nucleus (in μm^2^). Additionally, there is a probability $\;P_{\nu }\left(d\right)$, depending on dose, that exactly ν ions are crossing the nucleus for this certain dose *d*:(27)$$P_{\nu }\left(d\right)=\frac{N_{ion}^{\nu }}{\nu !}\;e^{-N_{ion}}$$

Considering a cell nucleus with diameter D_n_, the average length of an ion’s crossing through it is:(28)Xnc¯=π×Dn4

The number of lethal DNA lesions in the cell nucleus per ion $\;Y_{c\;}$ will be:(29)$$Y_{c}=\frac{dN_{l}}{d\zeta }\times \bar{X_{nc}}\times \overset{\infty }{\underset{\nu =1}{\sum }}\nu P_{\nu }\left(d\right)$$

And finally, the cell survival probability П*_surv_* is:(30)$$\underset{surv}{П}=e^{-Y_{c}}$$

Thereupon, we may calculate the survival fraction of any cell irradiated with anion, expressing it as a function of dose *d*, and then compare it to experimental data of the same cell irradiated in aerobic or hypoxic conditions. We did so for a large number of mammalian cells (for both the different values of the coefficient γ). Knowing the nuclear characteristics of every cell (cross-sectional area, diameter of the nucleus, and genome size) we can obtain its survival curves after irradiation with an ion of specific LET. 

The formula and values used above refer to the case of γ = 0.0001. In the alternative case of γ = 0.01 that we introduce innovatively in this paper, the only thing we must change is the value of the probability that an electron incident on this elemental target has to induce an SSB, $\;\Gamma_{SSB}$ = 1 [87].

## 5. Conclusions

We describe an improved adaptation of the MSA model to predict the initial complex DNA damage to cells irradiated with ions and especially heavy ions, the MR model. In our model, the starting point of the calculations was the singly-differential cross sections which determine the main characteristics of secondary electrons. The latter plays a key role in assessing cellular damage at the DNA scale. The next step was to define an elemental target of DNA and calculate damage to it by secondary electrons and free radicals before reducing these results to the whole extent of the macromolecule in the cell nucleus. Overall, calculated survival curves for a broad spectrum of cell series and heavy ions were found to be in good agreement against experimental data (see Figure 1). Importantly, the present methodology uses only three input parameters (in human cells only two input parameters) without fitting to the experimental data. Another advantage of this model is that it produces very satisfactory results in assessing complex DNA damage and cell survival generally for asynchronous irradiated cell populations. We optimized in a more logical reasoning, the quantification of free radicals in the close area around the ion’s trajectory based on reliable parameter values of the modern corresponding bibliography. We also applied our model to many different heavy ions which are being studied during the last decades on the basis for example of optimization of ion-beam therapy for medical treatment or radiation protection during space travelling. For the latter, studying radiation-induced biological damage by such heavy ions would aid scientific research to assess more accurately hazards of human exposure to cosmic radiation during space missions and cancer risk.

We calculated the RBE, a key quantity for the description of radiation biological effects induced by ions on living matter, which, despite its complexity and dependence on several parameters, is still a fundamental factor for determining the quality of an oncological radiation treatment. On the same basis, we calculated the OER which helps us to compare the radiobiological effects of ions to those at different aerobic and hypoxic conditions of the irradiated cell. 

The advantages of the MRM can be summarized as follows: (1) it uses only a limited number of input parameters (three in total) of which two are dependent on each applied cell. Although this is not uncommon for other more established empirical models like LQ, RMF still the parameterization number for MRM is considered low (but more complicated and difficult to apply) and plus (2) it uses no fitting to experimental data (the only fitting done in the whole model is that of the probability $\Gamma_{SSB\;}$of an electron incident on a DNA target to induce an SSB) and (3) it predicts cell survival curves for a broad spectrum of cells irradiated with many different heavy ions having a broad range of LET values (even for ^56^Fe ions which exist in cosmic radiation). Our model is not cell-cycle-phase specific but predicts survival very well for asynchronous cell cultures (Figure 1). Last but not least, (4) it is extendable to new physical quantities and theories (e.g., the shock waves, still undiscovered, but theoretically predicted in high LET radiation), to biological phenomena and new discoveries in radiobiology (the damage caused in mitochondria and other cell organelles during the irradiation). 

The current limitations are that no repair mechanistic model has been incorporated yet and therefore ‘shouldered’ survival curves cannot be predicted, but this is something primarily for X-rays as well-known and discussed above. Another major limitation when it comes to radiation therapy (RT), is that MRM cannot be applied yet to fractionation treatment protocols and in general incorporate human tissue effects after RT. 

## Figures and Tables

**Figure 1 molecules-26-00840-f001:**
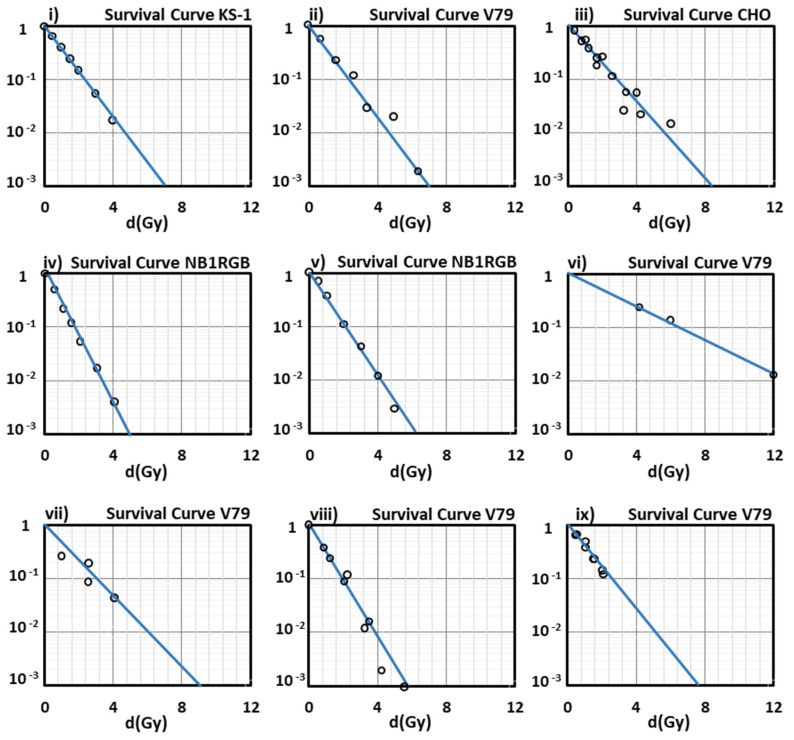
Survival curves (**i**–**ix**) for various heavy ions and a wide range LET values, obtained by this proposed model for asynchronous cell populations. In each diagram the dots represent the corresponding experimental data. Very good agreement is ascertained. (**i**) Survival curve for KS-1 cells (human brain glioblastoma) irradiated with ^12^C of LET = 13.3 keV/μm (for γ = 0.0001) [54]. (**ii**) V79 cells (Chinese hamster) irradiated with ^12^C of LET = 359.5 keV/μm (for γ = 0.0001) in hypoxic conditions [55]. (**iii**) CHO cells (Chinese hamster ovary cells) irradiated with ^16^O of energy *T* = 1424 MeV, (for γ = 0.0001) [20]. (**iv**) NB1RGB cells (normal human skin fibroblast) irradiated with ^28^Si of LET = 59 keV/μm (for γ = 0.01) [56]. (**v**) NB1RGB cells (normal human skin fibroblast) irradiated with ^20^Ne of LET = 45 keV/μm (for γ = 0.01) [56]. (**vi**) V79 cells (Chinese hamster) irradiated with ^56^Fe of LET = 2106 keV/μm (for γ = 0.0001) [57]. (**vii**) V79 cells (Chinese hamster) irradiated with ^40^Ar of energy *T* = 17138.5 MeV (for γ = 0.01) [58]. (**viii**) V79 cells (Chinese hamster) irradiated with ^14^N of LET = 125 keV/μm (for γ = 0.0001) [59]. (**ix**) V79 cells (Chinese hamster) irradiated with ^11^Β of LET = 110 keV/μm (for γ = 0.01) [60].

**Figure 2 molecules-26-00840-f002:**
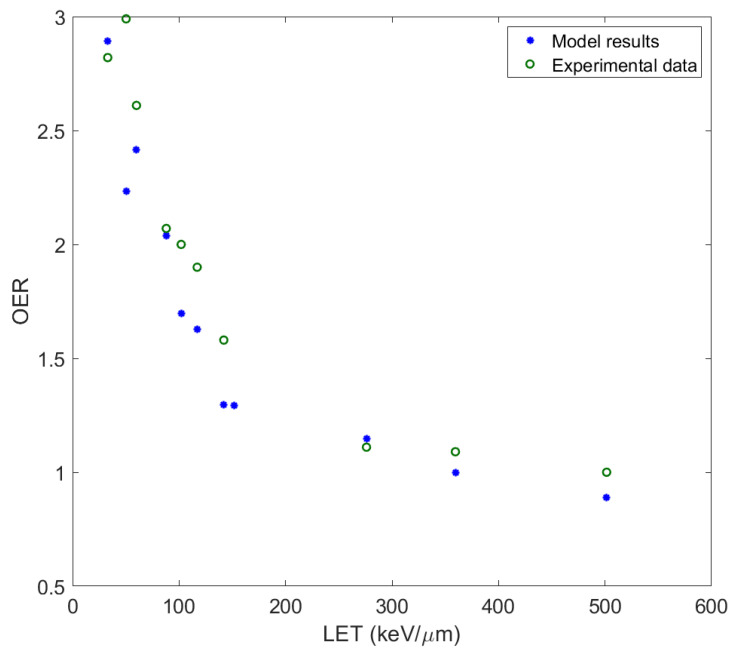
Distribution of the Oxygen Enhancement Ratio (OER) at the 10% survival level for V79 cells exposed to ^12^C (for γ = 0.0001) for different Linear Energy Transfer (LET) values. Our results are depicted with blue dots while the experimental data are depicted by the green open dots [53].

**Figure 3 molecules-26-00840-f003:**
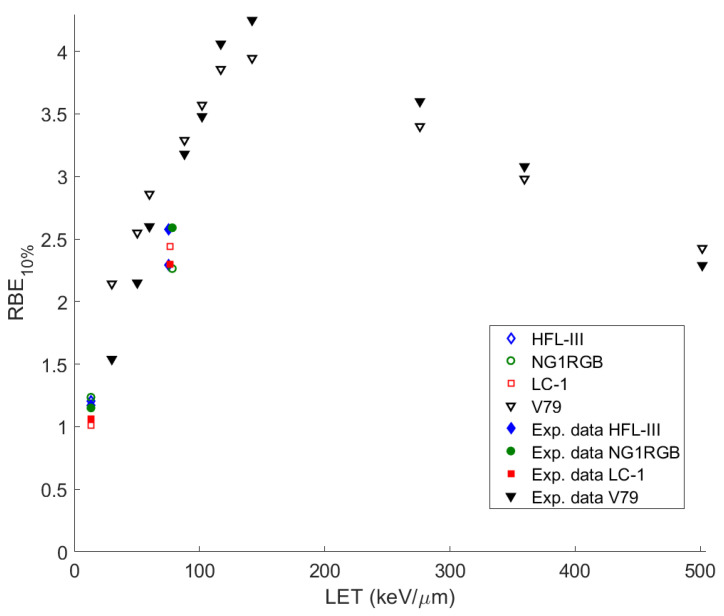
Distribution of the relative biological effectiveness (RBE) at the 10% survival level for V79, HFL-III, NG1RGB and LC-1 cells exposed to ^12^C ion irradiation (for γ = 0.0001) (our results are depicted with open symbols while the experimental data are the filled symbols) [53,54].

**Figure 4 molecules-26-00840-f004:**
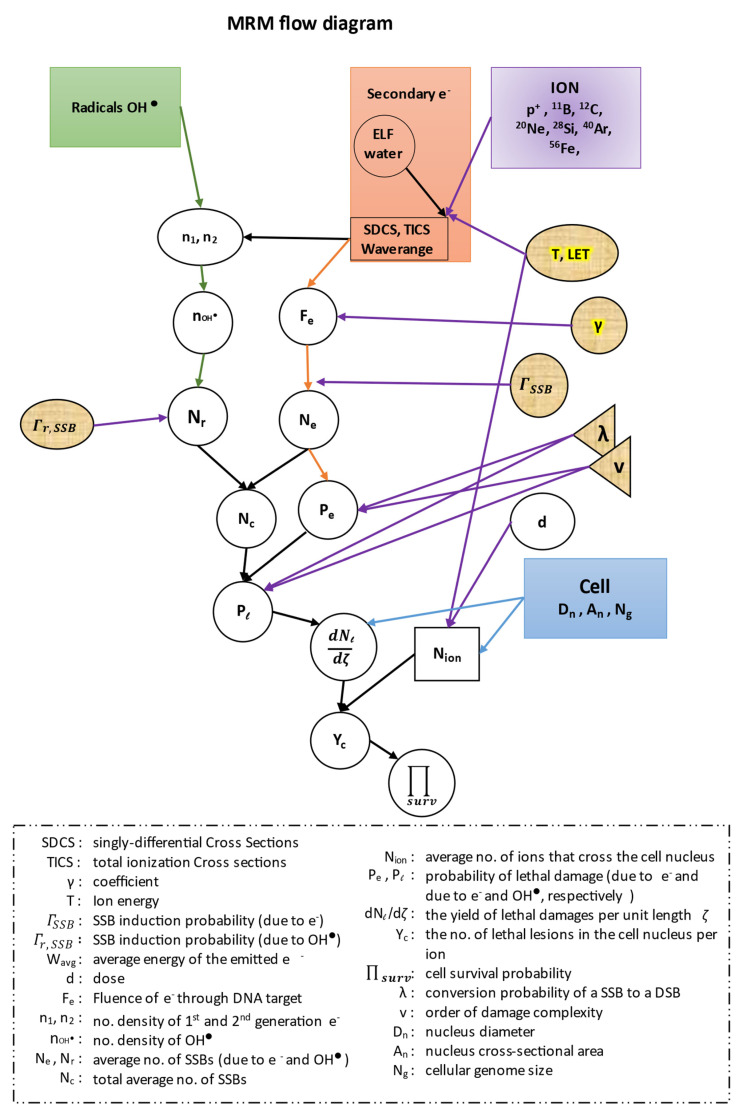
Flow diagram of the calculations in the mathematical radiobiological model (MRM) model.

**Figure 5 molecules-26-00840-f005:**
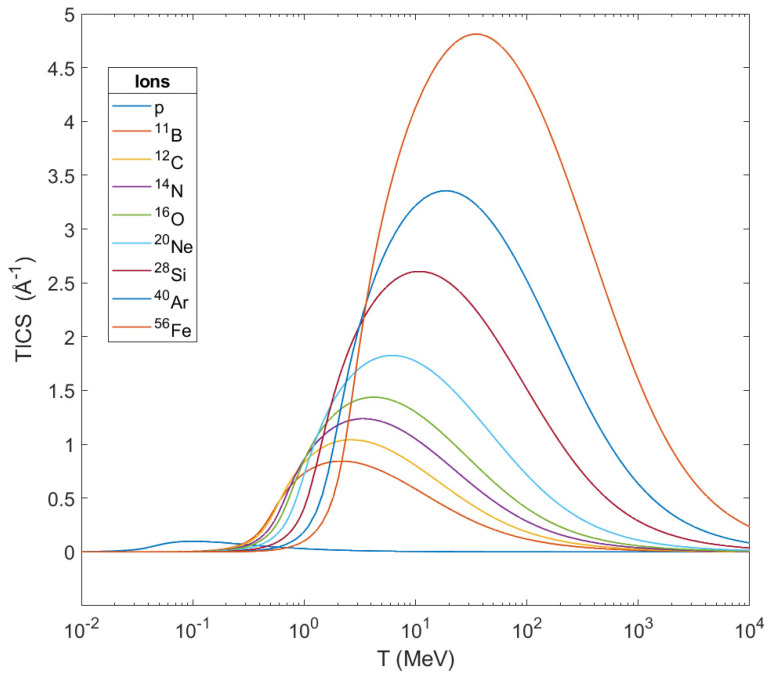
Macroscopic total ionization cross sections (TICS) in (Å^−^^1^) for protons, ^11^B, ^12^C, ^14^N, ^16^O, ^20^Ne, ^28^Si, ^40^Ar and ^56^Fe in liquid water as a biological material, as a function of ion’s kinetic energy T  (in MeV).

**Figure 6 molecules-26-00840-f006:**
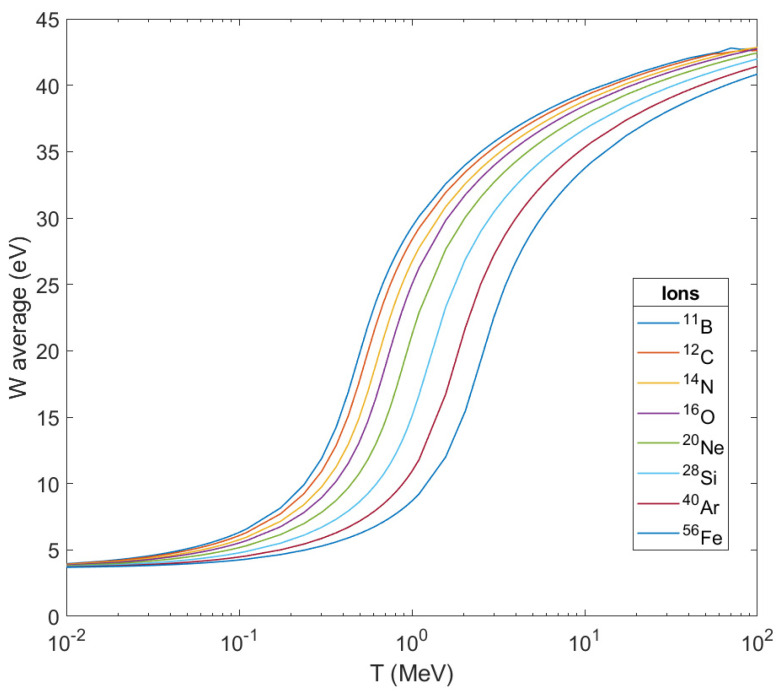
Average kinetic energy of secondary electrons (in eV) emitted in liquid water for the ions ^11^B, ^12^C, ^14^N, ^16^O, ^20^Ne, ^28^Si, ^40^Ar and ^56^Fe as projectiles, as a function of the ion’s kinetic energy T (in MeV).

**Figure 7 molecules-26-00840-f007:**
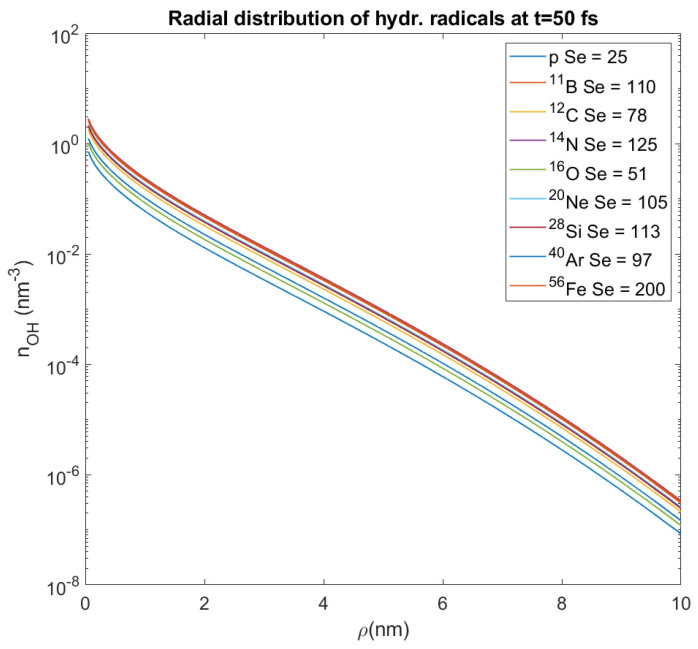
The initial radial distribution of OH^•^
$n_{ΟΗ•}\left(\rho \right)\;$ (at time t≃50 fs) for different heavy ions with different random values of stopping power S_e_ (keV/μm).

## Data Availability

Data are available upon logical request from the corresponding author.

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
