# Peer review of "A Mathematical Radiobiological Model (MRM) to Predict Complex DNA Damage and Cell Survival for Ionizing Particle Radiations of Varying Quality"

_molecules, 2021, doi:10.3390/molecules26040840_

Round 1

Reviewer 1 Report

Manuscript by Kalospyros et al. entitled “A Multiscale Radiobiological Model (MRM) to predict complex DNA damage and cell survival for ionizing particle radiations of varying quality” describes results of a computational modeling of DNA double-strand break formation and cell survival after exposure to heavy ions using a multiscale approach (MSA). Using this approach, the authors define multiple parameters for their mathematical model based on many considerations of the processes such as formation and travel of electrons along the ion tracks, cross-sections, probabilities of interaction with targets in DNA, of formation of complex DSBs as lethal damage, etc. They compare then the results generated by the model with the experimental results: survival curves, oxygen enhancement effect and relative biological effectiveness values (RBE), showing overall superior concordance between the calculated and experimental data.

The modeling approach seems to be indeed innovative and comprehensive and the study would be interesting for the radiobiological scientific community. Unfortunately, this Reviewer lacks strong expertise to evaluate the quality and validity of mathematical modeling used. However, biological reasoning and justification used in the course of the creation of the model seemed appropriate, with one important exception constituting one of the major comments.

Major comments

  1. The choice of water as an exclusive media for the modeling of the formation of electrons along the track of a heavy ion seems inappropriate. This is discussed in L238-249, without providing a single reference (e.g. reference is needed on L244). The statement in L245-247 is wrong. It has been shown that direct damage constitutes ~70% or a total damage to cells by particles of 2 MeV/um, equivalent to Fe-56 used in this study https://pubmed.ncbi.nlm.nih.gov/19267547/. It follows then that interaction of a particle with biomolecules, including DNA does play a significant role in the biological effect. It is therefore very strange that this arguably incorrect assumption resulted in survival curves that still in agreement with the experimental curves. Does this mean that the chosen parameters of the model perform well, but do not reflect well the biology of the process? Does this further mean that they were selected in order to concord with the experiment? These are important questions that need to be addressed/corrected by the authors.
  2. Partially similar comment regarding the missing role of DNA repair component. How come the lack of this component, correctly marked by the authors as an important one for future development of the proposed model, did not lead to discrepancy between model results and experiments? Obviously, cells had all kind of inducible repair mechanisms triggered and operational upon the experiments used for comparison with the model results. Same questions follows regarding the choice of the parameters to fit experiment.
  3. The choice of the conversion of SSB to DSB as 0.15 may be argued. Indeed a much higher ratio of SSB:DSN 23:1 has been shown (see e.g. https://pubmed.ncbi.nlm.nih.gov/23560631/ or multiple references can be easily found citing 1000 SSB and 50 DSB per Gy). Would the model perform well if this parameter is changed to 0.04 for example? Would this explain a bias as discussed in 1 and 2 above?

Other comments

  1. A flow diagram of the creation of the model would be very helpful
  2. Sections Methods and Results look very similar. In fact, no formulae are given in Methods (e.g. L364). Consider better aggregation of text between Methods and Results.
  3. Discussion needs to be modified:
    1. large portions of text in Outlook and future directions must be moved to Discussion. As is Discussion has only one reference.
    2. More should be said on the comparison of the model to other models (probably move text from the Introduction).
    3. Discuss limitations e.g. sensitivity of the model to changing parameters (SSB:DSB for one)
    4. Link inference to cancer to DNA repair, since tumorigenic events are improperly repaired DNA lesions; higher level tissue microenvironment and the immune system factors, etc.
    5. Discuss performance of the model at low doses and chronic exposures. Is protraction of exposure a factor (Yes, in this Reviewer’s opinion, but this can be acconted for by a lacking DNA repair component)
    6. Since the bystander effect was mentioned, inducible repair and radioadaptive responses should be mentioned and discussed too.
  4. L40-41: does such a difference really exist?
  5. L59: section 1.1. is not needed; check numbering of sections
  6. L63, L82, L245: requires reference. Some other statements throughout the manuscript also require references, please check
  7. Reference 41 is incomplete
  8. Please check – multiple typos (e.g. numbers breaking sentences in the abstract) must be corrected.
  9. The manuscript should be read and style corrected by a native English speaker

Author Response

Please see attachement. 

Reviewer 2 Report

The authors use in an appropriate manner the words "multiscale". In biophysics it is accepted to use multiscale to refer to models that go across molecular, cellular and tissue effects. Even at the cellular level it has not model of repair, no details on DNA structures, and does not describe shouldered survival curves or the transition from shouldered to exponential as ionization density is increased. Instead the so-called multiscale model is mostly a presentation of energy deposition.

The Introduction is way too long. I lost interest in the rehashing of many empirical models in the long introduction.

In the end the paper is an empirical model describing straight-line on log scale, ie exponential survival curves.  A model not mentioned that already did so 50 years ago was by Robert Katz which is never mentioned.  

A new empirical model describing V79 survival curves is not very important.

Author Response

RESPONSE TO THE REVIEWERS’ COMMENTS

            We sincerely thank the reviewers and the Editor for their constructive criticism. We have addressed all their comments and we have changed the text according to these comments as shown in our revised manuscript. Parts of our following responses to the Reviewers’ comments which have been included in the revised manuscript are printed in italics (followed by the referred lines of the revised manuscript). We also ask for the reviewers’ understanding that it was very difficult to reduce the Introduction part on different survival models for completeness, inclusiveness and allowance to compare our model with main models used in the literature.

REVIEWER 2

We kindly thank Reviewer 2 for his wise and targeted suggestions helping us improve clarity and potency of our work. Analytically:

  Comments:

  1. ‘A model not mentioned that already did so 50 years ago was by Robert Katz which is never mentioned’.

Response to comment a:

The amorphous track structure model which was introduced by R. Katz in the mid-60s for describing the response of biological systems to heavy ion irradiation [16-18] (L90-92) used the dose response for cell inactivation by γ-rays and correlated it with the radial dose distribution to predict the effects of ions. At that time, the knowledge of electron tracks was used for defining the properties of the ion ones represented in two dimensions (L93-94) The relation of the radial dose distribution from the ion’s path with the cell survival probability in that model was obtained based on the data for X-rays. That model also explained the changing shape of survival curves and the variation of radiosensitivity with LET (L98-99).

We had referred to the Amorphous Track Structure Theory (ATST) in L91-96 of our first manuscript, but we now make it clear about R. Katz’s model (L90-99).

  1. ‘The Introduction is way too long’

  Response to comment b:

We have decided to keep the extent of the introduction although too long, because we had to compare our model with the other models in order to show its advantages (as they described in the Section 5. ‘Conclusions’, see L889-901). Adding new parts to the other sections and reconstructing the first text in a new (suggested) layout in our revised manuscript, we feel that our extensive introduction with brief analysis of the most important radiobiological models, gives to the reader a synoptic and spherical knowledge of the most important radiobiological models used in radiation therapies. On the other hand, the parallel extension of the other sections of this revised version vanishes the former disanalogy of the Introduction in comparison to the other sections.

  1. ‘A new empirical model describing V79 survival curves is not very important’

   Response to comment c:

We have studied a large number of different cell lines in our model, the majority of which are a significant part of experiments that have been included and summarized in the PIDE database [105] and we demonstrate mostly the survival curves of the cell line V79 in order to observe also the behaviour of a typical extensively studied cell line exposed to irradiation with different ions and LET values (with a lot of experimental data in literature). Thus, we have chosen to demonstrate representatively five curves of the cell line V79, two of normal human cells (NB1RGB), one of cancer cell lines (KS-1) and one of non-human (CHO) mammalian cells (see Fig.1).

Reviewer 3 Report

The manuscript by Kalospyros at al. describes an interesting and useful mathematical MultiScale model  for the analysis of complex DNA damage and cell survival following exposure to high-LET heavy ions. One of the key advantages of this model is the ability to predict the effects of exposure as the function of Relative Biological Effectiveness (RBE) value of high-LET heavy ions. The manuscript is clearly written and presented. The results of this study should be of great interest to the readers of your journal. I am therefore happy to recommend it for publication in present form.

Author Response

We sincerely thank the reviewer for the very positive feedback and understating of our work. We have followed Editor’s and reviewers’ suggestion to further increase clarity and the understanding of our study. I hope we again satisfy the reviewer with the new and revised version.  

Round 2

Reviewer 1 Report

Thank you for answering all the comments and questions. This Reviewer has no further comments.

Author Response

We sincerely thank the reviewer for the positive assesment of our work and revisions. We do appreciate his/her time and effort to help us improve the manuscript. 

Reviewer 2 Report

Unfortunately very little was changed from the initial submission. 

The model is does not describe shoulders in cell survival curves and makes no attempt to do so. Therefore it would have no use for describing fractionation in therapy. 

The title of the manuscript is mis-leading since it is not a multi-scale model. It does not describe DNA damage repair or tissue controls or any aspect of tissues. It only describes radiation damage and a few aspects of initial chemistry. So it is not a Multi-scale model. 

The conclusion misses the mark since many models only use 2 or 3 parameters and are much simpler to apply, while also describing fractionation and shouldered survival curves. 

Author Response

We thank the reviewer for his/her additional comments and we now understand better the essence of his/her criticism. We read carefully all the additional comments and we revised the manuscript again extensively with several minor changes and highlighted also major ones as suggested by the reviewer.

We agree with the essence of the comments and we have tried to respond based on the following major ideas and empasize all these in the text like the limitations of the now called Mathematical Radiobiological Model (MRM) and not Multiscale as previously mentioned. More specifically:

1. We emphasize that our MRM model is currently only applicable to heavy ions and to dose-linear effects (no curvature), not repair mechanisms with no fractionation capability at least currently. We must say though that the discussion on the lack of specific mechanistic DNA repair components, it was already discussed in the 1st revision but now the limitation emphasized.  2. Despite being low-parameter number dependent, it is much more complicated (and difficult to apply) than the established empirical models. 3. Although we tend to agrree with the claim of being "multiscale" is questionable (in conventional terminology, multiscale models start from DNA level to predict tissue effects), for the history, we must say that our model scans for the intial molecular events up to cellular level (biological effect: survival). Nevertheless , we did changed EVERYWHERE this term and used for reference to the original theory and MSA model as the so-called 'Multiscale...' or 'MSA Model'.    We do hope that this re-revised manuscript will meet the Reviewer's high standards and cover his/her knowledgable comments and concerns.